# HSP90 Inhibition and Modulation of the Proteome: Therapeutical Implications for Idiopathic Pulmonary Fibrosis (IPF)

**DOI:** 10.3390/ijms21155286

**Published:** 2020-07-25

**Authors:** Ruben Manuel Luciano Colunga Biancatelli, Pavel Solopov, Betsy Gregory, John D. Catravas

**Affiliations:** 1Frank Reidy Research Center for Bioelectrics, Old Dominion University, Norfolk, VA 23508, USA; psolopov@odu.edu (P.S.); bgregory@odu.edu (B.G.); jcatrava@odu.edu (J.D.C.); 2Policlinico Umberto I, La Sapienza University of Rome, 00185 Rome, Italy; 3School of Medical Diagnostic & Translational Sciences, College of Health Sciences, Old Dominion University, Norfolk, VA 23508, USA

**Keywords:** Idiopathic Pulmonary Fibrosis, HSP90, HSP90 inhibitor, AUY-922, 17 AAG, Proteome, Proteomics, ERK, TGF-β, HSPome

## Abstract

Idiopathic Pulmonary fibrosis (IPF) is a catastrophic disease with poor outcomes and limited pharmacological approaches. Heat shock protein 90 (HSP90) has been recently involved in the wound-healing pathological response that leads to collagen deposition in patients with IPF and its inhibition represents an exciting drug target against the development of pulmonary fibrosis. Under physiological conditions, HSP90 guarantees proteostasis through the refolding of damaged proteins and the degradation of irreversibly damaged ones. Additionally, its inhibition, by specific HSP90 inhibitors (e.g., 17 AAG, 17 DAG, and AUY-922) has proven beneficial in different preclinical models of human disease. HSP90 inhibition modulates a complex subset of kinases and interferes with intracellular signaling pathways and proteome regulation. In this review, we evaluated the current evidence and rationale for the use of HSP90 inhibitors in the treatment of pulmonary fibrosis, discussed the intracellular pathways involved, described the limitations of the current understanding and provided insights for future research.

## 1. Introduction

Idiopathic Pulmonary Fibrosis (IPF) is a devastating disease, characterized by the progressive substitution of the lung parenchyma with a fibrotic scar. It is associated with poor prognosis and an estimated mean survival of 2–5 years from the time of diagnosis [1]. The pathophysiology of IPF is not completely understood; hypotheses for the principal causes of IPF include genetic predisposition, exaggerated immune response with fibroblast activation [2] and a defective mechanism in the wound-healing response to alveolar cell injury [3]. There are only two FDA-approved drugs for the treatment of IPF—Pirfenidone and Nintedanib [4]. Neither stops nor reverses the disease, but both slow disease progression and extend survival [5,6]. As of today, a cure is still missing and due to increasing trends in mortality over the last decades [7,8], there is an urgent need for the development of new pharmacological approaches capable of blocking disease progression and even restoring normal lung parenchyma. Proteomic studies have focused in identifying new biomarkers and the molecular pathology of IPF, utilizing peripheral blood, Bronchoalveolar Lavage Fluid (BALF) or parenchymal biopsies.

Heat shock proteins (HSPs) constitute a large family of co-chaperones classified by molecular weight and involved in the correct folding of a large number of “client proteins”. HSPs react to heat, hypoxia and oxidative stress, by assisting the assembly, stabilization, and translocation of oligomeric proteins [9]. Furthermore, when proteins are irreversibly damaged, HSPs promote their proteasomal degradation, avoiding the accumulation of deformed proteins in the cytoplasm [10]. The maintenance of a proper protein homeostasis (proteostasis) is crucial for cell survival, and is regulated by a complex and heterogeneous proteostasis network [11]. Through the refolding of misfolded proteins, segmentation of aggregated proteins and degradation of abnormal proteins, HSPs represent a crucial sophisticated machinery, which is activated by stress to preserve cell proteostasis [9,12,13]. While under physiological condition, HSPs act as guardians of the Proteome [14], their role in some diseases is uncertain. Pharmacological interventions that interfere within the HSP network have shown beneficial effects in cancer [15], neurodegenerative [16], and inflammatory diseases [17,18]. The rationale for these interventions is based on observations that persistent inflammation and damage to DNA can lead to abnormal activation of HSPs which could play a role in the pathophysiological mechanism of cell injury and disease progression.

HSP90, the most abundant of HSPs, is increased in patients with IPF [19] and has recently been investigated as a possible therapeutical target [20]. In this case, HSPs’ activation (phosphorylation) is suspected to be a pathological step that, activated by inflammation [21], promotes the synthesis and production of extracellular matrix and collagen. Accordingly, HSP90 inhibition represents an exciting approach that is collecting promising data in pre-clinical studies. Specifically, inhibition of HSP90 is considered effective in blocking the signaling pathway of Transforming Growth Factor-β (TGF- β), a crucial overexpressed profibrotic cytokine in Pulmonary Fibrosis (PF) (Figure 1) [22].

It is important to understand, however, that HSP90 interacts with a great number of proteins, kinases, and transcription factors and that its inhibition is followed by a strong modulation of the proteome and phosphoproteome, a large part of which is not completely understood. From this perspective, proteomic analysis of IPF represents one of the most promising methodologies in defining unknown pathways of IPF pathogenesis and in revealing how HSP90 inhibition may affect these pathways.

The aim of this review is to discuss the current understanding of HSP90′s role in the development of pulmonary fibrosis, the current proteomic understanding of IPF, evaluate available preclinical data and describe how inhibition of HSP90 could be a therapeutical target for patients with IPF.

## 2. Heat Shock Protein 90 Structure

HSP90 is one of the most expressed heat shock proteins, found in both *bacteria* and *eukaryotes* [23]. It plays an important role in the stress response to environmental insults (heat, hypoxia, and oxidative stress) as it mediates the correct folding and stabilization of several proteins, guaranteeing their function and promoting cell survival. HSP90 is a highly conserved ATP-dependent molecule composed by an N-Terminal ATP-binding Domain (NTD), a middle domain (MD), and a C-Terminal dimerization Domain (CTD). N- and M- domains are connected via a flexible linker of over 60 residues in length, which is important for HSP90 eukaryotic function, but is absent in bacterial and mitochondrial isoforms (Figure 2) [24].

The NTD is conserved among HSPs and shares homology with the ATPase/kinase GHKL (Gyrase, HSP90, Histidine Kinase, MutL) superfamily [25]. It presents an ATP-binding site which is 15 Å (1.5 nm) deep and cleaves ATP into ADP + P. This region is the principal binding site of HSP0 inhibitors (e.g., geldanamycin and radicicol) and currently under intense study for its therapeutic implications [26]. The MD is made up of three regions: a three-layer α-β-α sandwich, a three-turn α-helix with irregular loops and a six-turn α-helix. It has been suggested that the binding to Aha1, in a highly conserved tyrosine (Y313 of Hsp90α) of the MD, is responsible for the regulation of the conformational changes of HSP90, that modulate its intrinsic hydroxylating activity [27]. Accordingly, the enzymatic activity of HSP90 is also related to Arg-32, a key coupling element responsible for communication across HSP90 domains [28]. The third domain, the CTD interacts with co-chaperones like cyclophilin-40, PP5, stress-induced phosphoprotein 1 (Sti/Hop) and immunophilins FKBP51-52 through a tetracopeptide repeat (TPR) motif recognition site expressed at the end of the domain [29].

HSP90 chaperones form homodimers through the binding of the N-domain acquiring a V shape-dimer with ATP-depending conformational shifts [30]. This complex, in the ATP-bound state assumes a closed N-terminal domain, whose mechanism has not been completely understood [31]. However, it is clear that when HSP90 is phosphorylated it assumes a stronger chaperone activity, required during stress situations. The inhibitors, which have showed much higher affinity for the phosphorylated and pathological isoforms of HSP90 [32,33], bind the ATP-binding site of the NTD preventing ATP hydrolysis and reducing HSP90 chaperone activity [34]. It is important to note that the C-terminal domain of HSP90 presents an alternative ATP-binding pocket, which guarantees a minimal chaperone activity when the N-terminal binding pocket is occupied or inhibited [35].

## 3. HSP90 “Guardian” of the Proteome

HSPs represent a sophisticated protein quality-control network, which assist protein folding during assembly, and selectively degrade irreversibly damaged proteins. HSP90, the most studied and abundant of HSPs, is crucial for maturation of signaling proteins involved in cell division and development, such as steroid hormone receptors, kinases and key oncogenic proteins like the tumor suppressor p53 [36,37]. HSP90, together with HSP70 and other co-chaperones, promote the late-stage folding and maturation of more than 400 client proteins [37], including kinases, transcription factors, and E3 ubiquitin ligase [38]. In an attempt to define the human substrates that interact with HSP90, Lindquist et al. carried a quantitative analysis of HSP90-interactions and discovered that HSP90 forms complexes with 60% of human kinases, 30% of ubiquitin ligases, and ~7% of transcriptional factors [38]. The large number of interactions with the kinome appears to be dependent on HSP90′s cochaperone CDC37, as shown by the reduction in HSP90/kinase complex formation after CDC37 knockdown and, at the same time, suggesting CDC37 as a highly specialized cochaperone adaptor for kinases [39,40]. Upon HSP90 inhibition, HSP90-client kinases are redirected through degradation or accumulation.

Professor Didier Picard greatly contributed to the overall understanding of the HSP90 interactome, collecting the results of different studies and creating an online platform to help researcher in the general understanding of this complex network (www.hsp90.org [41]). However, as HSP90 not only regulates the proper function of several proteins but also modulates transcriptional factor, the definition of the pathways affected by HSP90 modulation are challenging and complex to define.

During stress, the Heat Shock Response (HSR) regulates the cytoplasmic proteostasis response, through the transcription of stress genes and the de novo synthesis of heat shock proteins in order to guarantee cell survival, adaptation to circulating hormones and protection of proteins from environmental insults [42,43]. Besides, HSPs direct, through HSP90′s leading role, a cytoprotective effect on cells, if cells are exposed to persistent inflammation and DNA damage, HSP90 could promote the exaggerate downstream signaling of inflammation promoting cell transformation and endorsing the folding of a tremendous amount of stress proteins, which end up accumulating.

The “HSPome”, defined as the part of the proteome affected by HSPs activity and modulation, as we previously mentioned, is constituted by an enormous number of functional proteins, transcriptional factors and ubiquitinases, whose expression could be regulated by the use of inhibitors resulting in profound changes in cell morphology and functionality, especially in response to stress.

In cancer, HSP90 regulates a high number of proteins involved in cellular signaling and tumor promotion, making HSP90 inhibitors a therapeutic approach with great potential [44]. Similarly, due to the vast difference of IPF proteome and HSPome compared to controls, inhibition of HSP90 is a promising intervention which could impede the profound changes in protein expression observed in late stage of the disease.

## 4. Proteomic Analysis in Idiopathic Pulmonary Fibrosis

Current proteomic understanding of IPF is derived from peripheral blood, bronchoalveolar lavage fluid (BALF), and lung tissue analysis. Patients with IPF exhibit variable rates of disease progression and for this reason many studies have focused on finding new biomarkers or indicators of therapeutic response to monitor these patients [45]. Proteomic profiling plays an important role in biomarker discovery [46] and preliminary studies have shown how patients with IPF display a unique peripheral blood proteome [47,48]. The IPF-PRO registry, a multicenter cohort study, analyzed over 1300 proteins in peripheral blood of patients with IPF and compared them to healthy controls, identifying nine proteins that accurately distinguished patients with IPF from controls. The circulating proteome differed for apolipoprotein A-1 (APOA1), complement C1r subcomponent, intracellular adhesion molecule 5 (ICAM5), C-C motif chemokine 18, 14-3-3 protein sigma (SFN), sonic hedgehog protein, oxidized low-density lipoprotein receptor 1, matrix metalloproteinase 3 (MMP3), macrophage-capping protein, and heat shock protein 90 beta-1 (HSP90β1) [49] (Table 1).

Even though the profile of circulating proteins differs from lung proteins, it can still provide insight into biological pathways linked to a persistent overexpression of inflammatory markers, adhesion molecules, complement proteins and HSPs.

BALF proteomic analysis of patients with IPF, compared to matched healthy controls, displayed an increased number of eosinophil and neutrophil proteins (eosinophil cationic protein, eosinophil lysophospholipase, and metalloproteinase 8 (MMP8)), profibrotic cytokines (osteopontin, C-C motif chemokine 18 (CCL19), and various collagens [50]. Carleo et al. compared BALF of familial IPF vs. sporadic IPF, highlighting different protein patterns in immune response (polymeric immunoglobulin receptors, and α-1-B-glycoprotein), coagulation (fibrinogen-γ, α1-antitrypsin, complement C3), wound response, oxidative stress (isocitrate, dehydrogenase, and peroxiredoxin 1) and ion homeostasis (ceruloplasmin, serotransferrin, and hemopexin) [51]. Furthermore, their findings on surfactant-associated protein A2 (SPA2) supported the IPF pathogenic theory of unresolved endoplasmic reticulum (ER) stress, SFTPA2 isoform mutation and accumulation, proteostasis disturbance and prolonged Unfolded Protein Response (UPR) signaling leading to type 2 alveolar-epithelial cells (AECII) apoptosis or senescence [57,58,59].

Furthermore, isoforms of Receptor for Advanced Glycation End-products (RAGE)—whose activity is related to activation of kinases (MAPK) and transcriptional factors (NF-κB) [60]—were analyzed in the BALF of IPF, COPD, and healthy subjects and revealed that RAGE (especially the endogenous soluble form (esRAGE)) is an important pro-fibrotic marker [52].

Isobaric tag for relative and absolute quantitation (iTRAQ) combined with liquid chromatography-tandem mass spectrometry (LC-MS/MS) was used to analyze proteomic changes in lung tissue of IPF patients and controls [53]. From more than 4000 proteins analyzed, 600 were founded altered (455 upregulated and 207 downregulated) confirming previous results on extracellular matrix overexpression and defining new involved proteins. Among these, several collagens, glycoproteins, proteoglycans, and secretory and regulatory factor matrisomes were identified and validated by Western blotting, confirming the role of collagen 1, cathepsin B, AGR-2, galectin-7, HSP90α, and HSP90β [53]. In another study, functional proteomics was used to analyze differences in four diseases (idiopathic pulmonary fibrosis, sarcoidosis, systemic sclerosis, and pulmonary Langerhans sclerosis). A network was observed linking alpha-1-antitrypsin, alpha-1-antychymotripsin (SEPRINA3), glutathione S transferase P1 (GSTP1) to 14-3-3 epsilon, its regulatory action on transcription factors, including heat shock factor 1 (HSF-1), NF-AT4, and *C-myc*, as well as inhibition of protein kinases [54].

In a murine model of bleomycin-induced PF, the proteome and secretome were studied at different time points in order to define fibrotic and regenerative stages [56]. Two unexpected proteins showed a significant increase in the fibrotic stage—Emilin-2 and Collagen-XXVIII—which were incorporated in the extracellular matrix (ECM) matrix and in the peribronchial and perivascular layers. A group of extracellular niche proteins and laminins were also identified which reacted to injury and whose expression is usually restricted to embryonic development. Emilin-2 was believed to bind Wnt1 in the extracellular space and promote the alveolar extrinsic apoptotic pathway [61]. Differently, in the recovery phase, Nrf2—a reactive oxygen species (ROS) transcriptional factor- increased at day 14 and 28, accompanying the recovery phase of bleomycin-induced fibrosis, as previously suggested [62]. Finally, proteomic study of IPF lungs and controls revealed a profound upregulation of HSPs including HSP90α, HSP90β, and mitochondrial HSP60, whose expression was associated with their role in the UPR and the activation of mitochondrial apoptosis pathways [55].

Taken together, these data strongly suggest that IPF could be classified as a disease with “unbalanced” proteostasis, whose main affected proteins could function as new biomarkers for disease severity. This data also indicates that HSP90 should be considered as a potential target for new therapeutic approaches.

## 5. HSP90 Pathways in Pulmonary Fibrosis

There is a rising amount of literature suggesting that HSP90 plays an important role in fibrogenesis [63] and that the observed high levels of HSP90 represent a pathological step required for disease progression (Table 2). IPF patients display high levels of HSP90 in plasma, whose expression is correlated with IPF severity, and fibroblast of patients with IPF secrete an increased amount of HSP90 [64]. Immunohistochemistry studies revealed the presence of both HSP90α and HSP90β in the pulmonary interstitium of IPF lungs with increased expression in myofibroblasts and fibroblast foci of IPF samples, compared to controls. Similarly, Western blot analysis showed significant upregulation of HSP90β in cultured Interstitial Lung Fibroblasts (ILFBs) from patients with IPF compared to controls [65]. Mechanical stretch of rat fibrotic lungs ex vivo provoked the release of HSP90 in the BALF [64].

HSP90α is able to promote the phosphorylation of Protein kinase B (AKT) in Thr308, P38, and extracellular signal–regulated kinase (ERK) signaling pathways, which are known downstream non-Smad-pathways of TGFβ1 signaling and participate in the development of pulmonary fibrosis [66]. Indeed, HSP90 has been shown to regulate ERK directly via dissociation of the ERK-HSP90-CDC37 complex [70] and indirectly by negatively inhibiting Raf metabolism and its downstream cascade [71]. Furthermore, HSP90 plays a critical role in TGFβ function by stabilizing its signaling cascade and receptors [63,72]. At the same time HSP90 regulates nuclear localization of Smad, influencing and modulating the Smad-dependent signaling cascade of TGF-β [73]. TGF-β, the leading cytokine in IPF, is responsible for the epithelial-to-mesenchymal transformation (EMT) of epithelial cells into a myofibroblastic phenotype that is related to the transformation of lung structure and the production of extracellular matrix (ECM) [74], thus, interference with its complex signaling cascade represents a reasonable therapeutic approach for IPF, supported by a strong mechanistic rationale. Changes in lungs parenchymal architecture are among the most obvious pathological signs of pulmonary fibrosis. Typical changes include diffuse alveolar edema and septal thickening, type II pneumocytes hyperplasia, damage to lung structure and accumulation of airspace fibrin and collagen [75,76]. HSP90 is overexpressed in lungs from IPF patients and high-magnification micrographs show that HSP90 is localized in both the cytosol and nucleus of fibroblasts [19]. Additionally, in a preliminary study, we observed increased HSP90 immunoreactivity in pneumocytes, airway epithelium, and fibroblast foci in areas of active fibrosis in rabbits with chemical-induced PF (Figure 3).

Strong expression of both HSP90α and HSP90β was observed in abnormal bronchiolar structures overlying fibroblast foci, as well as in hyperplastic bronchioles of IPF lungs. Hyperplastic type-II alveolar epithelial cells (AECII) near areas of dense fibrosis displayed robust immunostaining of HSP90β in IPF lungs compared to healthy donors, whereas HSP90α was absent in AECII. Smooth muscle cells located in the wall of bronchioles showed expression of HSP90α and α-SMA, but not of HSP90β [65]. Collagen type I secretion likely depends on the activity of HSP90 chaperones, even though such chaperone cannot directly engage nascent collagen molecules, but in addition to promoting TGF-β signaling, it could influence collagen-I secretion via interactions with cytosolic components of the secretory pathway [77]. Taken together these data suggest how HSP90 is involved in TGF-β receptor stabilization, its Smad-dependent and Smad-independent signaling cascade and could influence Collagen I secretion, representing an interesting target of new anti-fibrotic therapies (Figure 4).

HSP90 has also been found increased in animal models of pulmonary fibrosis. In the bleomycin-induced mouse model, BALF and serum displayed high levels of HSP90α and this increase was related to fibrosis [66]. Activated HSP90, phospho-HSP90, was elevated in lung tissue of mice with hydrochloric acid (HCl)-induced pulmonary fibrosis at 10 and 30 days after acid instillation, compared to controls [68]. HSP90 tyrosine phosphorylation induced by 3-hydroxy-3-methylglutaryl-coenzyme A reductase inhibitors was reported to increase its association with endothelial nitric oxide (eNOS) [78], a marker for pulmonary hypertension and pulmonary fibrosis [79]. Similarly, we observed increased levels of phospho-HSP90 in a murine model of nitrogen mustard-induced pulmonary fibrosis [69].

## 6. Modulation of HSP90 by Molecular Inhibitors

HSP90 inhibitors have shown already efficacious results in cancer [80] and infectious diseases [81], in which their ability to decrease protein folding resulted in reduced virus assembly and decreased tumor development. A precise intervention, such as by HSP90 inhibitors, modulates and reduces protein trafficking and restores proteostasis by diminishing HSP90 chaperone activity.

HSP90 inhibitors strongly decrease HSP90 interactions with kinases [38]. Sharma et al. defined the action of HSP90 inhibitors, 17-AAG and 17-DMAG, on the proteome and phosphoproteome, revealing that the blockade of the chaperone leads to more downregulated than upregulated proteins. Specifically, HSP90 inhibitors modulated ARAF, AKT, CDK4, MET, and PDK1 affecting principally protein kinase activity, as 34% of kinases were reduced and only 6% of them were upregulated [82]. In fact, HSP90 client kinases undergo several rounds of release and binding which promote the functional stability of folded proteins. However, when a HSP90 inhibitor is applied, HSP90′s ATPase activity is reduced and the reloading of clients into the chaperone is diminished [83]. These precise intervention redirects kinase clients into one of two paths: degradation or aggregation. Furthermore, HSP90 inhibition is not limited only to depletion of the proteome and phosphoproteome but also to the loss of activity of those signaling pathways formed by HSP90-dependent proteins [84]. This reduction cannot just be considered as a proteomic difference but as a metabolic redirection whose effects strongly differ from untreated cells. Schumacher et al. [85] showed how geldanamycin, a HSP90 inhibitor, downregulated 119 peptides, including integrin, estrogen receptor, MAPK, JAK/STAT, PPAR, and NF-κB as well as other cell cycle-regulatory and ubiquitin-mediated pathways. Another HSP90 inhibitor, IPI-504—a more water-soluble analog of geldanamycin—showed to redirect pancreatic cancer cell proteome from cell growth, maintenance, and transport to cellular metabolism, organization, and biogenesis [86]. The use of HSP90 inhibitors was also studied in HS68 fibroblasts and SW480, U2OS, and A549 cancer cells. The effect on the kinome, after HSP90 inhibitors application showed downregulation in 70% kinases of Hs68 cells (144 quantified kinases), including MAPK and TGF-β signaling pathways [87]. 1G6-D7, another HSP90α inhibitor, attenuated the severity of the fibrotic process in a murine model of Bleomycin-induced pulmonary fibrosis by lowering the circulating levels of HSP90α [66]. NVP-AUY922, a highly potent HSP90 inhibitor with improved bioavailability and aqueous solubility compared to first generation HSP90 inhibitors [88], successfully inhibited both HSP90α and HSP90β with similar median inhibition concentration (IC50) values [89]. It is widely known that the mitogen-activated protein kinase (MAPK)/MAPK kinase (MEK)/extracellular signal–regulated kinase (ERK) signaling cascade is a major pathway controlling cellular processes associated with fibrogenesis [90]. The expression of total MEK, p-MEK, total ERK, and also the phosphorylation of EGFR, a typical HSP90 client protein was found to be reduced by AUY-922 [91]. Our previous studies have shown that AUY922 can depress phosphorylation of ERK, interfering with TGF-β signaling, thereby preventing the development of HCL and Nitrogen Mustard-induced pulmonary fibrosis [21,92,93]. The use of HSP90 inhibitors regulates TGF-β receptor stabilization, interferes with Smad and non-Smad (Raf, P-ERK) TGF-β signaling cascade, activates transcription factors and therefore decreases epithelial-to-mesenchymal transformation (EMT) and reduces production of pro-fibrotic mediators and ECM. Furthermore, as HSP90 inhibitors were shown to modulate several MAP kinases involved in inflammation, cell proliferation and fibrogenesis, further studies are needed to define in particular these pathways in IPF.

By modulating the HSPome, HSP90 inhibitors affect complex intracellular networks involved in the development of pulmonary fibrosis and offer strong rationale as useful therapeutical interventions directing cells through protein degradation.

## 7. Current Limitation and Future Research

IPF is complex disease with unclear mechanisms of pathogenesis. Proteomic analysis together with genomic studies represent a promising approach towards defining unknown pathways of genetic predisposition and proteome unbalance in IPF. In recent years, several new molecules have been implicated in its pathogenesis, creating new potential biomarkers and therapeutical targets. However, it is important to note that every cell type presents its own proteome and some of the present data arise from mixed cell populations. In order to characterize cell-to-cell differences, it would be ideal to isolate either from BALF, lung tissue or peripheral blood each distinct cell type and carry its proteomic evaluation. This would greatly contribute to a further description of the role that each cell, either pneumocytes, endothelial or immune cells, play during the development of the disease.

Limitations also arise from the analysis of HSP90ome. HSP90 is involved in the stress response and consequently expression and activity levels are affected by experimental and environmental conditions that need to be considered when comparing different studies.

Further, HSP90 exists in multiple isoforms (e.g., α, β, αβ…) that have a distinct role in cellular metabolism, as some function in the cytosol, others in the nucleus, and others in the mitochondria, thus establishing distinct interactive networks whose understanding is far from clear [94]. Notably, a large section of the current literature has only partially defined these differences and due to study limitations, the understanding of HSP90′s nuclear isoforms is still limited.

Other disadvantages are related to the techniques used to study the relationship between HSP90 function and the proteome. Current technology is unable to distinguish between direct or indirect interactions between the chaperone and other proteins. In addition, defining the client proteins (HSPome) of HSP90 and how HSP90 inhibitors change their expression remains challenging. Large proteomic analyses describe changes in proteins expressed in relatively high concentration while HSP90 modulates many signaling and transcription factors which, are expressed at low levels in cells, but are characterized by a strong metabolic activity.

Further issues arise when comparing the main proteomic studies on HSP90 inhibitors. Thus, results from studies by Song [86], Maloney [95], Schumacher [85], and Sharma [82] differ significantly among them, with just a few molecules identified in common. This suggest that HSP90 inhibitors act differently on diverse cell lines, exerting unique cellular responses and underscoring again the importance of a cell-specific proteomic evaluation in IPF.

Even though more studies are required, proteomic studies from IPF patients and laboratory analysis of HSP90 inhibitors provide a strong mechanistic rationale for their investigation in IPF, justifying further experimentation in preclinical studies to better identify their anti-fibrotic effects, optimal dose strategy, and potential side effects.

## 8. Conclusions

IPF is a disease characterized by unbalance of the proteome. HSP90 plays an important role in the pathogenesis of the fibrotic process in the lung. Controlled interventions, as with HSP90 inhibitors, represent an exciting approach for the modulation of the proteome and phosphoproteome that has been shown to result in reduction of pro-fibrotic markers, ECM proteins and improved lung mechanics in several pre-clinical studies. Furthermore, HSP90 inhibitors critically interfere with Smad and non-Smad intracellular signaling of TGF-β. As the modulation of HSP90 results in changes in multiple pathways, further studies are necessary to define in detail how HSP90 inhibitors exert their beneficial effect in IPF, the optimal dose strategy, and potential side effects.

## Figures and Tables

**Figure 1 ijms-21-05286-f001:**
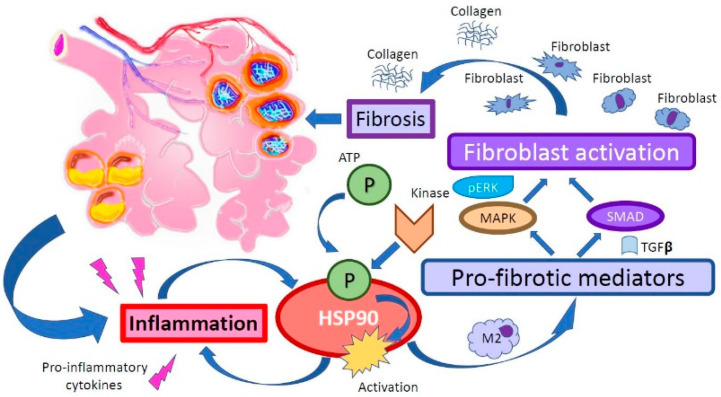
Heat shock protein 90 (HSP90) plays a role in the pro-fibrotic process in the lung that follows an inflammatory stimulus. Furthermore, it is involved in TGF-β signaling, fibroblast activation, and deposition of collagen.

**Figure 2 ijms-21-05286-f002:**
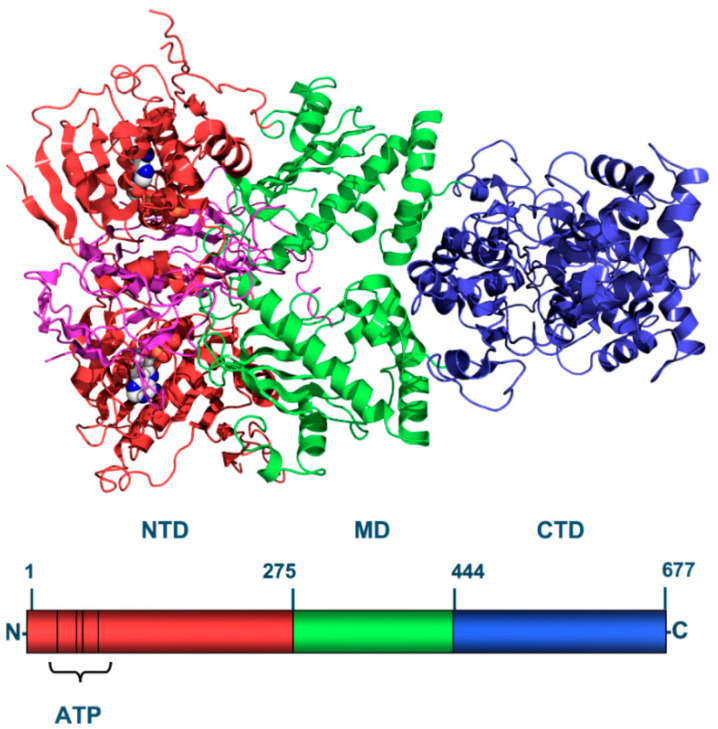
Primary structure of the yeast HSP90. The N-Terminal-Domain (NTD-red) is a highly conserved domain among HSPs and contains the ATP-binding pocket, target of many HSP90 inhibitors. The Middle-Domain (MD-green) is divided into three regions (a 3-layer α–β–α sandwich, a 3-turn α-helix and irregular loops and a 6-turn α-helix) and it is involved in client and substrate binding that increase ATPase activity (Aha1, Hch1). The C-terminal Domain (CTD-blue) possesses a moderate alternative ATP-binding site that become available when the N-terminal pocket is occupied.

**Figure 3 ijms-21-05286-f003:**
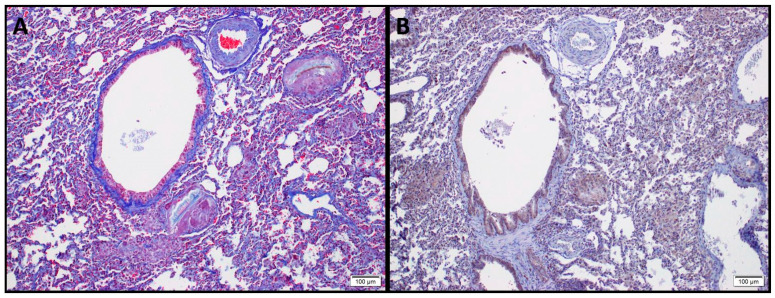
Sections from rabbit lungs 60 days after intratracheal instillation of 0.1 *N*–hydrochloric acid. (**A**) Masson’s Trichrome staining depicting fibrotic lesions, loss of alveolar architecture, peribronchial, and perivascular collagen deposition. (**B**) HSP90β immunohistostaining of the same section displaying upregulation of HSP90β (in brown) within the fibrotic tissue and in peribronchial regions.

**Figure 4 ijms-21-05286-f004:**
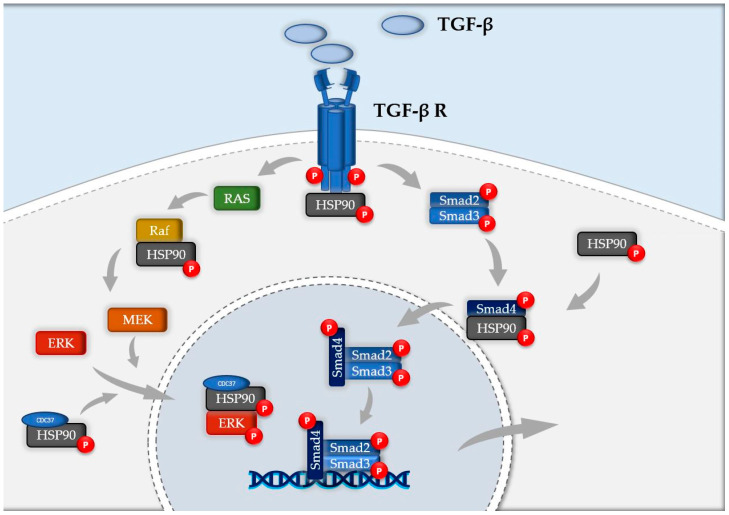
Schematic representation of HSP90 involvement in TGF-β signaling cascade. HSP90 plays a crucial role at various levels of the pathogenic pathway of IPF. It stabilizes TGF-β receptor, negatively regulates Raf, preserve ERK from degradation with its binding with HSP90-CDC37 complex and modulates nuclear localization of phospho-Smad4.

**Table 1 ijms-21-05286-t001:** Proteomic analysis in studies on Idiopathic Pulmonary Fibrosis (IPF).

Study	Species	Sample	Increased Expression
Todd et al. [49]	Human	Peripheral blood	APOA1, complement C1r subcomponent, ICAM5, CC-motif chemokine 18, 14-3-3 SFN, sonic hedgehog protein, Oxidized low density lipoprotein. Receptor 1, MMP3, macrophage capping protein, HSP90β-1.
Foster et al. [50]	Human	BALF	Eosinophil cationic protein, eosinophil lysophospholipase, MMP8, osteopontin, CC-motif chemokine 18 and collagens.
Carleo et al. [51]	Human	BALF	Polymeric immunoglobulin receptors, α-1-B-glycoprotein, fibrinogen-γ, α-1-antitrypsin, complement C3, isocitrate, dehydrogenase, peroxiredoxin 1, ceruloplasmin, serotransferrin, SPA2 and hemopexin.
Ohlmeier et al. [52]	Human	BALF	esRAGE
Tian et al. [53]	Human	Lung	Collagen 1, cathepsin B, AGR-2, galectin-7, HSP90α, HSP90β
Landi et al. [54]	Human	Lung	Alpha-1-antitripsin, SEPRINA3, GSTP1, 14-3-3, HSF-1, NF-AT4, C-Myc
Korfei et al. [55]	Human	Lung	HSP90α, HSP90β, HSP60
Schiller et al. [56]	mice	Lung	Emylin-2, Collagen-XXVIII, Wnt1

Summarized data from different proteomic studies on IPF. APOA1: Apolipoprotein A-1, ICAM-5: intracellular adhesion molecule 5; SFN 14-3-3 protein Sigma; MMP3: Matrix MetalloProteinase 3; HSP90β-1: Heat Shock Protein 90β-1; MMP8: Matrix MetalloProteinase 8; SPA2: Surfactant Associated Protein A2; esRAGE: endogenous soluble form of Receptor for Advanced Glycation End-products; AGR-2: Anterior Gradient Protein 2; HSP90α: Heat Shock Protein α; SEPRINA3: alpha-1-antychymotripsin; GSTP1: Glutathione S Transferase P1; HSF-1: Heat Shock Factor 1; NF-AT4: Nuclear Factor of Activated T cell.

**Table 2 ijms-21-05286-t002:** Expression levels of HSP90 and its isoforms in clinical and preclinical studies of IPF.

Study	Species	Etiology	Isoforms	Sample	Fold-Change
Korfei et al. [55]	Human	IPF	HSP90 α	Lung	>2.3
			HSP90 β	Lung	>4.8
Sontake et al. [19]	Human	IPF	HSP90	Lung	>1.5
Sibinska et al. [65]	Human	IPF	HSP90 α	Lung	>1.5
	Human	IPF	HSP90 β	Lung	>2.3
	mice	Bleomycin	HSP90 α	Lung	>2.0
	mice	Bleomycin	HSP90 β	Lung	>2.5
Hangming et al. [66]	mice	Bleomycin	HSP90 α	BALF	>9
				serum	>1.8
Bellaye et al. [67]	Human	IPF	HSP90 α	serum	>2.0
	Human	IPF	HSP90 β	serum	no change
	rat	Mechanical stretch	eHSP90	tissue	increased
	mice	Mechanical stretch	HSP90	BALF	increased
Marinova et al. [68]	mice	Hydrochloric acid	P-HSP90	Lung	>2.1
Solopov et al. [69]	mice	Nitrogen Mustard	P-HSP90	Lung	>2.0

Table 2 collates significant evidence from different clinical and preclinical studies of IPF, were HSP90 was found increased (fold-change expression levels) compared to healthy controls (*p* < 0.05). IPF: Idiopathic pulmonary fibrosis; HSP90: Heat shock Protein 90; P-HSP90: Phosphorylated Heat Shock Protein 90; eHSP90—extracellular Heat Shock Protein 90.

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
