# Peer review of "HSP90 Inhibition and Modulation of the Proteome: Therapeutical Implications for Idiopathic Pulmonary Fibrosis (IPF)"

_ijms, 2020, doi:10.3390/ijms21155286_

Round 1

Reviewer 1 Report

This is a nice review article on actual hot topic, which is worth for publishing in International Journal of Molecular Sciences. There are only few minor comments listed below:

  1. «HSP90, the most abundant of HSPs, is increased in patients with IPF [19]» - How many times has the HSPs’ concentration increased, compared to healthy volunteers? Is it statistically significant? It would be great to make a table with the findings.
  2. «In cancer, HSP90 regulates a high number of proteins involved in cellular signaling and tumor promotion, making HSP90 inhibitors a therapeutic approach with great potential [44]» - Are there any differences in cases of IPF and cancer?
  3. In paragraph 4 it would be good to add the table summarizing proteomics results in different studies (peripheral blood, bronchoalveolar lavage fluid, lung tissue).
  4. «HSP90 is involved in the stress response and consequently expression and activity levels are affected by experimental and environmental conditions, that need to be considered when comparing different studies» - Are there any ways to solve the problem: standardization of experimental conditions, reducing environmental impact? Are there any investigations that tried to reduce the limitations?

Author Response

  1. HSP90 isoforms were found significantly increased in several studies. Following reviewer comment, we developed a table summarizing the results from different studies in either preclinical or clinical experiments.
  2. IPF and cancer are two different diseases which however are characterized by broad changes in proteome expression. HSP90, acting on several pathways find its rationale in both as in cancer it modulates many oncoproteins and cell cycle regulators, while in IPF it interferes with proteins involved in epithelial to mesenchymal transformation and the production of extracellular matrix. Furthermore, preclinical studies suggest that the dose of HSP90 inhibitors needed to block the fibrotic process is different from that used in cancer cells.
  3. We added a table summarizing proteomics results from different studies.
  4. We thank the reviewer for their valuable comment and interest in the topic. The standardization of experimental condition and the use of a similar protocol among laboratories is clearly the right path to choose. However, many difficulties arise as researchers are used to work with specific reagents, different types of samples and heterogenic study designs which sometimes can’t be modified and standardized. We believe, however, that underlining the problem is the first step to find a solution and, maybe in few years, this would lead the way to create a committee for the development of guidelines and protocols for HSP90 analysis in IPF.

Reviewer 2 Report

This mini-review article by Ruben et al. summarizes the latest advances on the role of HSP90 and the use of its inhibitors to modulate its proteome to mitigate pulmonary fibrosis. There are several strengths to this review. The paper was well-written with excellent quality figures and points summarized are relevant to human IPF disease.

Minor Comment:

The authors should consider highlighting the findings from recently published studies on potential downstream targets (proteins or genes) of HSP90 isoforms and 17-AAG in mediating multiple pro-fibrotic functions of fibroblasts including proliferation, migration, myofibroblast transformation and ECM production (Page 8).

Author Response

We thank reviewer for their congratulation and comments. We added few sentences on paragraph 6. Moreover, we believe that we discussed in detail HSP90 role in IPF in paragraph 5 using the best available data. However, comprehensive mechanisms on how HSP90 inhibitors modulate the proliferative and pro-fibrotic response in IPF is scarce. We reviewed the few pathways that have shown significant results (P-ERK, SMAD, TGF-B, Raf), yet data on other pathways is missing and only continuing the research on this topic will add valuable information about alternative intracellular networks affected by HSP90 inhibitors.